# Optimization of Coffee Oil Extraction from Defective Beans Using a Supercritical Carbon Dioxide Technique: Its Effect on Volatile Aroma Components

**DOI:** 10.3390/foods12132515

**Published:** 2023-06-28

**Authors:** Wasin Pattaraprachyakul, Ruengwit Sawangkeaw, Somkiat Ngamprasertsith, Inthawoot Suppavorasatit

**Affiliations:** 1Department of Food Technology, Faculty of Science, Chulalongkorn University, Phayathai RD., Wangmai, Pathumwan, Bangkok 10330, Thailand; wasin_095@outlook.com; 2The Institute of Biotechnology and Genetic Engineering, Chulalongkorn University, 254 Institute Building 3, Phayathai RD., Wangmai, Pathumwan, Bangkok 10330, Thailand; rueangwit.s@chula.ac.th; 3Fuels Research Center, Department of Chemical Technology, Faculty of Science, Chulalongkorn University, Bangkok 10330, Thailand; somkiat.n@chula.ac.th; 4Flavor Science and Functional Ingredients Research Unit, Chulalongkorn University, Phayathai RD., Wangmai, Pathumwan, Bangkok 10330, Thailand

**Keywords:** defective coffee beans, supercritical fluid extraction, coffee oil, response surface methodology, volatile aroma components

## Abstract

Defective green coffee beans are typically discarded due to their negative impacts on coffee qualities compared to normal beans. However, there are some types of defective beans that can cause volatile aroma compounds after roasting similar to those produced by normal beans. This study aimed to optimize conditions for coffee oil extraction by supercritical carbon dioxide using the response surface methodology (RSM). Furthermore, the investigation assessed the aroma-active compounds and sensory quality in extracted coffee oil. Thus, operational temperatures (33.2–66.8 °C), pressure (10–30 MPa) and ethanol (g) to roasted coffee (g) ratio (0.25:1–1.5:1) were optimized for coffee oil extraction. As a result, different oil yields with different key volatile aroma compounds concentrations were obtained and it was found that the optimum conditions for extraction were a temperature of 50 °C, pressure of 30 MPa, and ethanol (g) to roasted coffee (g) ratio of 1:1 to obtain 6.50% (*w*/*w*) coffee oil yield. Key volatile aroma compounds, including furfuryl alcohol, 5-methyl furfural, 2,5-dimethylpyrazine, 4-vinylguaiacol, furfuryl acetate, 3-ethyl-2,5-dimethylpyrazine, thiazole, 1-furfurylpyrrole, pyridine, 2,3-butanediol, and 3-methyl-1,2-cyclopentanedione which contributed to the most preferable burnt, sweet, bready, chocolate-like, and roasted flavors, were quantified. Overall, the results suggested that coffee oil extracted from defective beans could be potentially used as a flavoring agent.

## 1. Introduction

Coffee consumption is widely prevalent in the world, making it one of the most popular beverages. However, this substantial output of coffee beans is accompanied by a significant occurrence of defective beans, accounting for approximately 20 percent of the total production or an estimated 0.61 million tons annually [1]. Due to their effect on sensory attributes, these defective beans are commonly rejected by the market [1]. Major defects include black, sour, brown, broken, and immature beans, which occurred from improper bean formation within the coffee cherries or inadequate harvesting and processing practices, such as strip-picking [1,2]. It has been observed that the formation of off-aromas/flavors is associated with the black, sour, and brown beans, while broken and immature defective beans do not contribute to this phenomenon [3]. Furthermore, Oliveira et al. [1] found that the proximate compositions of defective and non-defective arabica green coffee beans did not differ significantly, except in terms of ash content. Consequently, the volatile aroma compounds produced in roasted coffee beans derived from broken and immature beans may exhibit similarities to those generated by non-defective beans [4].

As mentioned earlier, off-aroma/flavor is not associated with broken and immature beans. The broken bean can be obtained from mature fruit, which is broken by mistake during processing. For the immature fruit, it has been reported that compositions in the beans were not much different from mature ones [5]. Therefore, coffee aroma/flavor can be formed from the similar precursors to the non-defective beans [6]. The chemical reactions that occur during roasting and produce important volatile aroma/flavor compounds include: Maillard reactions (non-enzymatic browning), phenolic acid and carotenoid degradation, Strecker degradation, degradation of trigonelline, chlorogenic acids, quinic acid, pigments, and lipids, as well as reactions between other intermediate products [7,8].

Coffee oil is recognized as an impact carrier of the aroma and flavor found in roasted coffee, encompassing a substantial portion of the aromatic compounds that contribute to the delightful coffee flavor [9]. In order to determine aroma and flavor of extracted oils, gas chromatograph-mass spectrometry can be applied to identify and quantify volatile compounds in the oils [10,11,12]. Additionally, some authors have identified over 30 volatile compounds in extracted coffee oil, such as 2,5-dimethylpyrazine, furanmethanol, 2-ethyl-3-methylpyrazine, difurfuryl ether, 2-furfurylfuran, 5-methyl-2-furancarboxaldehyde, furanmethanol acetate, 2-metoxy-4-vinylphenol, 2-furancarboxaldehyde, and 4-ethyl-2-methoxyphenol, which have the acceptable aroma associated with roasted coffee [10,11]. Moreover, with distinctive qualities, coffee oil finds extensive application as a flavoring agent in many food products, such as ice cream, beverage, and instant coffee, as well as serving as a natural ingredient for cosmetics [9,10]. To enhance the aroma and flavor of coffee products, such as roasted beans, canned coffee, and instant coffee, manufacturers use flavors derived from the oil of roasted coffee beans in the products [13]. In addition, upon heating, a large proportion of the volatile compounds in the beans tend to evaporate or degrade, thereby affecting the flavor quality of the coffee [14]. Thus, the application of flavoring agents can effectively preserve the desired flavor attributes of coffee products during storage [10,14].

Flavor profiles of extracted coffee oils can vary depending on different extraction processes [15]. The supercritical carbon dioxide (CO_2_) extraction technique has been reported as one of the modern methods that has not been subjected to high heat, hydro distillation, and toxic solvents. In addition, both non-volatile and volatile compounds can be recovered using this technique, which differ from extracted oils from hydro-distillation and solvent-extraction techniques [16]. In the case of solvent extraction, it is important to note that the solvent extraction method was employed with high temperatures (>50–110 °C) and extended extraction times (>2 h), which may potentially generate artifacts and thus affect the aroma of the oils. It was reported that using dichloromethane could provide coffee oil with old coffee, burnt, and phenolic attributes. This observation can be explained by the heat-induced degradation during the extraction process and a higher extraction efficiency for hydrophobic volatile aroma compounds as an effect of dichloromethane [15,17,18].

Supercritical fluid extraction employs temperatures and pressures to extract oil, utilizing the extraction principle above the critical point [19]. One significant advantage of using CO_2_ as a solvent in supercritical fluid extraction is that no residual solvent remains in the extracted oil since it changes to a gas at atmospheric pressure after extraction [20]. The critical temperature for carbon dioxide is 31.3 °C, and this method operates at a relatively low-temperature range, thus reducing the risk of heat-induced decomposition of the extract [19,20]. Although solvent extraction poses a risk of leaving chemical residues in the oil, it is still highly efficient when combined with supercritical fluid extraction, capable of large-scale extraction of flavoring compounds depending on the properties of the solvent and extraction conditions [21].

Araújo et al. [22] conducted a study in which lipids were extracted from spent coffee grounds using CO_2_ with ethanol under various conditions. These conditions included different pressures (10, 15, and 20 MPa), temperatures (40, 60, and 80 °C), and ethanol to sample ratios (0.25:1, 0.5:1, 1:1, and 2:1 *w*/*w*). The purpose of the study was to demonstrate the technical feasibility of employing green technology in this process. The highest yield, amounting to 14.14%, was achieved when the extraction was carried out at 20 MPa, 80 °C, and an ethanol to sample ratio of 2:1 (*w*/*w*) for 25 min. Moreover, they investigated the effect of adding ethanol as a co-solvent to the system. Specifically, they utilized an ethanol- to-sample mass ratio of 0.5:1 (*w*/*w*) at 80 °C and 20 MPa. This resulted in an extraction yield of 9.13% under the same conditions. Furthermore, it was been observed that the supercritical CO_2_ extraction method predominantly yields volatile compounds in coffee oil belonging to the furan and pyrazine families such as furanmethanol, 2-ethyl-3-methylpyrazine, and difurfuryl ether, which contribute to the desirable nutty, roasty, and smoky aroma associated with roasted coffee beans [11].

As mentioned earlier, the yield of extracted oil can be affected by several factors, including pressure, temperature, methods, and solvent used. To the best of our knowledge, there are few studies related to the yield and the acceptance of aroma oil extracted from roasted defective coffee beans. Moreover, a few have addressed the volatile aroma components of oil extracted from roasted defective coffee beans. The optimizing of coffee oil extraction from defective beans can be considered as potential benefits in terms of waste reduction in the coffee bean production process and efficiency of utilization of resources. Thus, the objectives of this study were to optimize extraction conditions of coffee oil using the response surface methodology and to identify volatile aroma compounds in the extracted oil using the direct solvent extraction coupled with the gas chromatography–mass spectrometry/olfactometry (GC-MS/O) technique.

## 2. Materials and Methods

### 2.1. Materials

Defective (broken and immature) green arabica coffee beans (*Coffea arabica*) were purchased from MTT Organic Coffee Farm in Chiang Mai province, Thailand. The bean samples were stored in sealed HDPE laminated jute bags and kept in a dry area at ambient temperature. All chemicals, reagents, and solvents were analytical grade and used as purchased.

### 2.2. Sample Preparation

Roasted coffee beans were prepared according to the method of Franca et al. [23] with some modification. Green defective beans (20 g) were placed in a metal tray lined with a parchment paper baking sheet and put into a preheated oven (SO6102TS, Smeg S.p.A., Guastalla, Italy) at 200 °C. After 13 min roasting, the roasted beans were placed into another metal tray and cooled down at room temperature for 15 min. After that, the coffee silver skin was removed through sieving and then the beans were divided and placed into odorless laminated bags (MOPP/VMPET/PE) equipped with CO_2_ degassing valves. The beans were kept at 25 ± 2 °C in a dark place for not more than 14 days before extraction.

### 2.3. Supercritical Carbon Dioxide Extraction with Co-Solvent

The roasted coffee beans were ground to a particle size between 250–500 μm (40 mesh). The experimental setup consisted of a cooling bath (Heto Lab Equipment, Allerød, Denmark) and a thermostatic bath (Heto Lab Equipment, Allerød, Denmark) to maintain the temperature of a high-pressure pump, an extractor vessel with internal volume of 40 cm^3^ coupled to a thermostatic vessel, which was used to keep a constant extraction temperature. The solvent mass flow of 5 g/min was controlled using a high-pressure pump. The extractions were conducted under controlled conditions of temperature and pressure, specifically at 33.2–66.8 °C and at 10–30 MPa. The ethanol (g) to roasted coffee (g) ratio was in a range of 0.25:1–1.5:1. The central composite design (CCD) was employed in the experiment and specific 17 conditions were measured, as shown in Table 1. A constant mass of ground coffee (20 g) was used in each experimental condition. To prevent the infiltration of solid particles into the piping system, the inlet and outlet of the extractor were lined with ceramic fiber. Then the extracted oil was evaporated by the vacuum evaporator at 40 °C for 30 min. Yields were calculated using Equation (1) [22].
(1)$$\mathrm{Yield}\;\left(\%w/w\right)=\frac{\mathrm{Weight}\;\mathrm{of}\;\mathrm{extract}\;\mathrm{oil}\;}{\mathrm{Weight}\;\mathrm{of}\;\mathrm{sample}\;\;}\times 100$$

### 2.4. Determination of Volatile Aroma Compounds in Coffee Oil

The direct solvent extraction (DSE) technique was modified from the method described by Pua et al. [24]. To carry out the extraction, extracted oil was placed in a 10 mL volumetric flask. Ten μL of 1000.467 μg/mL 2,4,6-trimethylpyridine was spiked into the flask containing a sample as an internal standard (i.s.). Dichloromethane was added to adjust the volume to 10 mL, then sealed and mixed vigorously. Water and impurities were removed from the sample using anhydrous sodium sulfate and the sample was further concentrated by N_2_ purge until the sample volume was reduced to approximately 200 µL. The concentrated samples were then stored in sample vials, which were sealed with a PTFE-coated silicone septum (Agilent, Santa Clara, CA, USA) and kept at −20 °C prior to analysis.

In order to determine volatile aroma compounds in the samples, 2 μL of the extract was injected in an injection port using the split mode (1:5) of an Agilent 7890B gas chromatograph (GC) equipped with a mass spectrometer (MS) and olfactometer (sniffing port, O). Temperature was programmed to hold at 40 °C for 5 min and then heated at a rate of 5 °C/min to a final temperature of 220 °C, and held for 5 min. Helium gas was used as a carrier gas at a constant flow rate of 2.0 mL/min. The DB-Wax column (30 m × 0.25 mm × 0.25 μm) (Woodbridge, VA, USA) was used as a stationary phase for this study. At the end of the column, the compounds were split into an MS and a sniffing port. The MS conditions were set as follows: Transfer line temperature, 250 °C; ionization voltage, 70 eV; mass range (scan mode), 35 to 350 amu. Concurrently, an experienced panelist was asked to sniff at the sniffing port. Only compounds detected by at least 2 out of 3 panelists were reported as volatile aroma compounds. In order to quantify the mass of the compounds, the method explained by Temthawee et. al. [25] was used, with some modification. In this approach, the response factor (*f_i_*) of MS analysis was employed for comparison purposes, utilizing an internal standard. The *f_i_* of each compound is defined as the inverse of the slope of a standard curve of peak area ratio (aroma compound/i.s.) versus mass ratio (aroma compound/i.s.) for an ascending series of mass ratios. To obtain the retention index (RI) of each compound, retention times of standard alkanes ranging from C10 to C40 (Sigma-Aldrich, St. Louis, MO, USA) were used for the calculation using the method described by Kulapichitr et al. [26]. The identification of volatile compounds was achieved through the comparison of their mass spectra (NIST 14.0 library), RI, and odor description. Compound quantification was determined against the peak area of each volatile to the i.s. with *f_i_* value (Equation (2)). In addition, the odor activity value (OAV) of each compound was also calculated following the published method [27]. Triplicate experiments were performed, and the mean values and standard deviations were reported.
(2)$$\mathrm{Relative}\;\mathrm{concentration}\;\left(µg/\mathrm{kg}\right)=\frac{\;(\frac{A_{V}}{A_{i.s.}}{)\;\times \;M}_{i.s.}}{M_{\mathrm{sample}}}\times f_{i}$$
where:

A_V_ = peak area of volatile compound

A_i.s._ = peak area of the internal standard

M_i.s._ = mass of the internal standard

M_sample_ = weight of sample

*f_i_* = response factor of volatile compound

### 2.5. Sensory Evaluation

An acceptance test conducted in this study was approved by the Research Ethics Review Committee for Research Involving Human Research Participants, Health Sciences Group, Chulalongkorn University (COA No. 037/66). Thirty panelists (20–37 years old, 11 males and 19 females) who regularly drink at least five cups of coffee per week were recruited to participate in this study. The study was conducted in booths illuminated with a red light to minimize interference with sensory perception. The extracted oil samples were diluted to a concentration of 20 mg/mL in medium-chain triglyceride (MCT) oil. Samples were served in 10 mL portions, placed in 75 mL amber glass sample vials, and sealed with plastic-lined caps. The samples were labeled with random 3-digit codes and randomly served to the panelists to evaluate aroma by giving a liking score using a 9-point hedonic scale [28].

### 2.6. Statistical Analysis

The experimental design and subsequent data analysis were carried out utilizing response surface methodology (RSM), with the Statistica^®^ software (Version 12.0, Stat Soft Inc., Tulsa, OK, USA). Through this methodology, second-order polynomial models were developed for the significant responses within the quadratic model, while the correlation coefficient (R^2^) and the F-test derived from the analysis of variance (ANOVA) were used as criteria for assessing significance. Furthermore, response surfaces and contour plots corresponding to the respective mathematical models (Equation (3)) were obtained, and significance was accepted at *p* ≤ 0.05. Two-way analysis of variance (ANOVA) and Duncan’s new multiple range tests (DNMRT) were employed to analyze differences in the mean values at a 95% confidence interval. These tests were used to determine the quality parameters of the extracted oil. The data were processed using IBM SPSS software (Version 19.0, SPSS Inc., Chicago, IL, USA). All measurements were performed in triplicate experiments.
(3)$$Y_{i}=\beta_{0}+\beta_{1}X_{1}+\beta_{2}X_{2}+\beta_{12}X_{1}X_{2}+\beta_{11}{X_{1}}^{2}+\beta_{22}{X_{2}}^{²}+\mathrm{error}$$
where Y represents the response variable, while β_0_ as the constant coefficient, β_i_ and β_j_ as the linear coefficients, β_ii_ and β_jj_ as the quadratic coefficients, β_ij_ as the linear-by-linear interaction coefficient, and X_1_ and X_2_ to represent the coded values of the independent variables.

## 3. Results and Discussion

Experimental conditions employed during the extraction of roasted coffee oil using supercritical CO_2_ are shown in Table 1. The table presents both coded and actual units of the variables, corresponding to the yield obtained for each experiment. From Table 1, the highest oil yield (7.68% *w*/*w*) was achieved at a pressure of 20 MPa, temperature of 33.2 °C, and an ethanol-to-sample ratio of 0.875:1 (*w*/*w*). These findings aligned with previous studies by Hurtado-Benavid et al. [11], who reported a yield of oil extracted from roasted coffee using supercritical CO_2_ at 22.5 MPa and a temperature of 36 °C was 5.32 ± 0.48% *w*/*w* (without ethanol). In addition, Oliveira et al. [19] also reported that the highest oil yield (7.75% *w*/*w*) was obtained from roasted non-defective bean with the supercritical CO_2_ technique at 27.5 MPa and a temperature of 42 °C without ethanol. However, it is noteworthy that despite increasing the ethanol ratio to 1.5:1, the highest yield was not achieved. This phenomenon may have happened because of mass transfer resistance caused by excessive ethanol in the extraction vessel, potentially impeding CO_2_ diffusion into the solvent-solids phase and resulting in reduction of extraction rates [29]. Additionally, the addition of ethanol as a co-solvent might affect the shifting of the critical point of the system to higher temperatures and pressures, thereby enhancing the solvent’s affinity for solubilizing highly polar compounds in the sample [30].

From response surface methodology, a mathematical model was fitted to explain % yield as dependent variable and which is shown as a second-order equation (Equation (4)).
Yield (%*w*/*w*) = 10.469 + 0.362X_1_ − 0.782X_2_ + 0.151X_3_ − 0.003X_1_X_2_ − 0.002X_1_X_3_ − 0.001X_2_X_3_ + 0.004X_1_^2^ + 0.009X_2_^2^ − 0.0001X_3_^2^
(4)

The coefficient of determination (R^2^) of % yield equation was 0.912, which means the model could explain 91.2% of total variation. The statistical significance of the factors influencing the yield is presented in Table 2. Among the factors examined, three factors were found to have a statistically significant impact (*p* < 0.05) on the yield, including pressure (X_1_), ethanol-to-sample ratio (X_3_), and the quadratic interaction of temperature (X_2_^2^). In addition, it was found that the main effect (B: temperature) of the model was not significant, thus the constant temperature at 50 °C was set as a constant value in Equation (4), and thus the adjusted model is shown in Equation (5).
Yield (%*w*/*w*) = −6.7089 + 0.228388X_1_ + 0.1X_3_ − 0.00209X_1_X_3_ + 0.004006X_1_^2^ − 0.00013X_3_^2^
(5)

The analysis using response surface methodology was employed and the response surface and contour plots are shown in Figure 1. It was found that extraction yield was increased when increasing both the ethanol-to-sample ratio (expressed as a percentage) and the pressure. This observation is inconsistent with the findings reported by Hurtado-Benavid et al. [11], who reported the positive effect of pressure on the solvent’s density, consequently enhancing the solubility of the oil. Additionally, the addition of ethanol as a co-solvent was known to improve the extraction yield due to its polar nature. Ethanol exhibited a stronger affinity for polar constituents within the extract, resulting in a higher output of extracted oil [21,31]. Moreover, increasing the pressure facilitates the penetration of the solvent into the solid matrix, thus promoting efficient contact between the solvent and solute [22].

When considering the response surface plots of the model (Figure 1a) incorporating the X_1_X_3_ equation, it implied that pressure and ethanol to sample ratio played a significant role in optimizing the extracted yield. Through the response contour plots (Figure 1b), which describes the region of interest represented by the dark red color, there is an optimized yield of >6%. Moreover, the contour plots predict points that correspond to specific yield conditions, providing valuable guidance for achieving an optimized response. In the results, the specific point characterized by Pressure (X_1_) = 30 MPa, Temperature (X_2_) = 50 °C, and ethanol to sample ratio (X_3_) = 1:1 stood out as it represented the minimum ethanol ratio and pressure necessary to attain a yield of at least 6%. In order to validate the optimized model, these conditions were re-extracted in triplicate experiments, and it was found that the observed extraction yield was 6.50%, which is higher than the predicted value of 6.00%. Thus, the chosen conditions were proved to be suitable for high extraction yield. 

Furthermore, Andrade et al. [32] investigated the optimization of oil extraction from coffee grounds using supercritical CO_2_. Their findings indicated that pressure levels between 10 MPa and 20 MPa provided higher extraction yields at lower temperatures since the density of the solvent was increased. Conversely, pressure levels between 20 MPa and 25 MPa resulted in higher yields at higher temperatures due to the higher vapor pressure of the solute. At pressures exceeding 25 MPa, lower temperatures led to higher yields due to the enhanced solvent density, highlighting the influence of pressure on solvent behavior. These findings were in agreement with that observation made by Araújo et al. [22] regarding the negative impact of temperature on the extraction yield. As temperature increased, the oil yield decreased due to the decrease in the density of the supercritical fluid, consequently reducing the oil’s solubility [22,33].

In addition, supercritical CO_2_ extraction influences the majority of non-polar compounds present in roasted coffee, which predominantly consist of lipids. These lipids encompass approximately 75% triacylglycerol, 19% total free and esterified diterpene alcohols, 5% total free and esterified sterols, and a small quantity of tocopherols [10]. During extraction, it is essential for the CO_2_ to reach a critical point within the extraction vessel to sustain the extraction process. Furthermore, the incorporation of ethanol within the extraction system serves to solubilize high polar compounds, consequently influencing the overall yield of the extracted mixture containing both lipids and volatile compounds. 

An analysis of volatile compounds using DSE and GC-MS/O techniques during preliminary experiments, five key volatile aroma compounds of coffee oil were identified, including furfuryl alcohol, 5-methyl furfural, 2,5-dimethylpyrazine, 4-vinylguaiacol, and furfuryl acetate, which are found in coffee oil extracted by the supercritical carbon dioxide technique [11]. 

In Table 3, the response factor (*f_i_*) of each key aroma compound was calculated. The equations obtained from each standard curve showed the R^2^ value at over 0.996 for all equations, indicating that they account for more than 99.6% of the total variation in the data. The *f_i_* for each compound in Table 3 was used in the calculation of relative concentration of key aroma compounds in coffee oil. For other volatile compounds quantitation, the *f_i_* = 1 was used in the calculation of their relative concentrations [25].

Table 4 provides the relative concentration of volatile aroma compounds and the average liking score for coffee oil under 17 different conditions (as same as shown in Table 1). Notably, exp. 11, which provided the highest coffee oil yield (7.6% *w*/*w*), exhibited a total concentration of key volatile aroma compounds at 1339.65 mg/kg. However, this concentration was significantly lower than other conditions, except exp 1. This finding suggested that the low concentration of total volatile compounds in exp. 11 may affect the perception of aroma and subsequently influence the liking score (6.50), which means panelists slightly to moderately like the aroma of the oil sample. Interestingly, it was found that the liking score observed for samples obtained from the exp 1 was 7.29, even though it contains similar overall key aroma compound content to samples from exp 11. It could be explained by the differences in each key compound in both samples [34]. Additionally, the highest yield (%*w*/*w*) of extracted oil was observed in the exp 11 (Table 1). However, the total key volatile aroma compounds at the same experimental treatment (exp 11) were significantly lower among all treatments (Table 4). This could show that key volatile aroma compounds concentrations were affected by dependent variables (pressure, temperature, and ethanol-to-sample ratio). In addition, the total concentrations of key volatile aroma compounds varied between approximately 1323 to 5413 mg/kg, which was not consistent with extracted oil yields shown in Table 1. Furthermore, it was observed from 17 experiments (Table 4) that the liking scores for coffee oil samples fell within the range of 6.50–7.47, indicating a preference among the panelists. This observation could show the potential application of carbon dioxide-ethanol as an extraction method to extract coffee oil from roasted defective coffee beans to meet consumer’s expectation. 

From the aforementioned validation of extraction conditions, the obtained coffee oil was analyzed for its key aroma compounds relative concentration and sensorial preference. The relative concentration of key aroma compounds is shown in Table 5. It was found that the concentration of key volatile aroma compounds fell within the range shown in Table 4. The compound with highest concentration was furfuryl alcohol, followed by 5-methyl furfural, 2,5-dimethylpyrazine, 4-vinylguaiacol, and furfuryl acetate, respectively. Additionally, the liking score of the extracted coffee oil was 7.51 ± 1.19, indicating a moderate to very high level of preference. 

In terms of volatile aroma compounds, it had been reported that the majority of the volatile aroma compounds which represented coffee aroma were liposoluble and can be extracted from the lipids of roasted coffee beans [10,35]. Sarrazin et al. [15] reported that the extraction techniques significantly impact the recovery of aroma compounds. They found that using solvent extractions by dichloromethane on medium roast coffee exhibited an old coffee-like aroma, which was not similar to what was extracted by compression or supercritical CO_2_ extraction.

Volatile aroma compounds of coffee oil extracted from the optimum condition were analyzed using DSE and GC-MS/O techniques. A total of 35 volatile compounds were positively identified. In order to know which compounds were really important in terms of aroma, olfactometry (O) using experienced panelists helped in identification. It was found that 23 volatile aroma compounds were positively identified, as shown in Table 6. The majority of the volatile aroma groups based on their concentrations (only compounds that peak areas could be identified, Appendix A) were furans (70.46%), pyrazines (7.12%), phenols (6.32%), pyrroles (5.68%), ketones and lactones (4.24%), alcohols (3.03%), pyridines (1.71%), and thiazoles (1.44%). Among these 23 compounds, there were 6 compounds indicated as “unknown” because these compounds were perceived by the human nose while sniffing, but the peak was not shown in the chromatogram. In fact, these compounds contributed aroma to the coffee oil; however, the compound concentrations were not high enough to be detected by machine. 

In addition to sniffing, identification of aroma active compounds was achieved by the calculation of odor activity value (OAV) of the compound. Generally, humans cannot perceive aroma from the compounds containing OAV ˂ 1 (compound concentration less than its odor threshold) [27]. Therefore, the aroma-active compounds found in the coffee oil sample were 3-ethyl-2,5-dimethylpyrazine, 4-vinylguaiacol, 2,5-dimethylpyrazine, furfuryl acetate, furfuryl alcohol, 2,3-butanediol, 5-methyl furfural, 1-furfurylpyrrole, 3-methyl-1,2-cyclopentanedione, 2-phenylethanol, furfural, 2-acetylfuran, 2-pyrrolecarboxaldehyde, 2-acetylpyrrole, pyridine, and thiazole. These compounds could be categorized into eight groups, which represent different odor qualities; pyrazines (burnt and nutty), phenols (phenolic and spicy), furans (fruity, bready, and sweet), pyrroles (musty and woody), alcohols (balsamic and sweet), ketones and lactones (fruity and sweet), pyridine (burnt), and thiazole (meaty). Our findings correspond to Hurtado-Benavid et al. [11], who found furfuryl alcohol, furfuryl acetate, 4-vinylguaiacol, 5-methyl furfural, and 3-methyl-1,2-cyclopentanedione as the major volatile compounds obtained from roasted coffee oil by supercritical CO_2_ extraction. It was reported that volatile compounds of the coffee belonged to the classes of hydrocarbons, alcohols, aldehydes, ketones, carboxylic acids, esters, pyrazines, pyrroles, pyridines, sulfur compounds, furans, furanones, phenols, and oxazoles. Moreover, furans and pyrazines were the most abundant classes of volatile compounds in coffee and most significant contributors to coffee flavor [8,37]. 

From Table 6, it was found that 3-ethyl-2,5-dimethylpyrazine has the highest OAV at 158,738, followed by 4-vinylguaiacol (10,178), 2,5-dimethylpyrazine (1932), furfuryl acetate (1208), and thiazole (1160), respectively. Additionally, it was found that another seven compounds contained OAVs ≥ 100, including furfuryl alcohol, 2,3-butanediol, 5-methyl furfural, 1-furfurylpyrrole, 3-methyl-1,2-cyclopentanedione, pyridine, and 2-phenylethanol. The other four compounds contained OAVs ≥ 1, including furfural and 2-acetylfuran, 2-pyrrolecarboxaldehyde, and 2-acetylpyrrole. Interestingly, the types of compound found in the present study were similar to key aroma compounds in brewed Arabica coffee reported by Kulapichitr et al. [38]; however, the compound concentrations and OAVs were different due to the difference in extraction methods.

The Maillard reaction is an important process to generate volatile aroma compounds, such as pyrazines, pyridines, and pyrroles in coffee, involving dehydration, fragmentation, and polymerization reactions during the coffee roasting process [7]. Furthermore, furfural can be formed through the oxidation of furfuryl alcohol, which results from the reaction between sugar and amino acids containing sulfur [8]. Among the volatile compounds identified, furans were found to be the most abundant group (Table 6). The results showed the presence of five furans, including furfuryl alcohol, 5-methyl furfural, furfuryl acetate, furfural, and 2-acetylfuran. Furfuryl acetate showed the highest OAV of 1208 among the furans group and it was associated with a fruity-like aroma. Moreover, furfuryl alcohol, 5-methyl furfural, furfural, and 2-acetylfuran could contribute to the burnt, sweet, nutty, and bready aroma of coffee oil.

Pyrazines were found to be the second most abundant class of compounds in the sample. These pyrazines are formed from the self-condensation and oxidation of α-aminoketones, which were formed during the Strecker degradation process [7,8]. Among pyrazines identified, 3-ethyl-2,5-dimethylpyrazine was found to be the predominant pyrazine with the highest OAV in coffee oil. Notably, this compound has previously been reported in roasted defective coffee beans, especially in immature beans [4]. Furthermore, 3-ethyl-2,5-dimethylpyrazine and 2,5-dimethylpyrazine could contribute to the burnt, sweet, nutty, and chocolate-like aroma of coffee oil. In addition, 4-vinylguaiacol had been reported due to the presence of a spicy phenolic aroma [24]. The formation of these phenolic compounds could have occurred through thermal degradation of chlorogenic acids, ferulic, caffeic, and quinic acids. The concentrations of these compounds could be directly influenced by the levels of organic acids present in the corresponding green coffee beans [39].

Pyrroles were reported that were caused by the fragmentation of 3-deoxyglucose during the Maillard reaction and Strecker degradation [7,8]. In particular, 1-furfurylpyrrole showed the highest OAV (307) in the pyrrole group and contributed a burnt aroma. Moreover, 2-acetylpyrrole and 2-pyrrolecarboxaldehyde (OAV = 1) were also found, which were associated with a musty, woody-like aroma, which refers to roasty and smoky attributes [7]. In addition, the degradation of trigonelline during coffee roasting could result in the formation of pyridine and nicotinic acid [7], which pyridine was reported to be associated with the aroma of aged-roasted coffee. Furthermore, some compounds of the pyridine class, such as 2-methylpyridine, have been reported as its contribution to the astringency of coffee [24]. In the present study, 52 mg/kg pyridine was found, and the OAV was 524. This pyridine contributed to the burnt aroma during sniffing.

Low molecular weight ketones are abundant in roasts which have experienced a decline in concentration during storage over time. Additionally, cyclic ketones were mainly found in roasted coffee beans like 3-hydroxy-2-methyl-4H-pyran-4-one and 3-methyl-1,2-cyclopentanedione and were associated with a burnt sugar-like aroma [8]. However, only 3-methyl-1,2-cyclopentanedione and acetoxyacetone were found in the present study. In addition, alcohol in the coffee, produced via the metabolic process of yeast, could interacted with fatty acids and amino acids to form esters that contribute to the fruity and floral aromas in the final product [8]. Furthermore, the presence of 2-phenylethanol could be primarily contributed from the activity of *Yarrowia lipolytica*, one of yeast species involved in the wet processing during fermentation [40]. Interestingly, 2,3-butanediol has been identified as a characterized marker compound for roasted defective beans [4]. In the present study, it has been identified at around 49 mg/kg, with an OAV of 516, which was associated with a balsamic-like and sweet aroma. In addition, 2-phenylethanol showed an OAV of 112 and it was associated with a floral-like aroma. 

The findings suggest that coffee oil could be potentially used as flavoring agents for many food products, such as beverages and bakeries. Previous studies have demonstrated that extracts obtained through the supercritical CO_2_ technique have the ability to improve the overall preference of the consumers [14,36]. Further investigation could be considered into the stability or shelf-life of the extracted coffee oil, and the application of the oil in flavoring and food industries.

## 4. Conclusions

Results have revealed that temperatures of 50 °C, pressure of 30 MPa and ethanol (g) to roasted coffee (g) ratio of 1:1 were optimum conditions for coffee oil extraction from roasted defective coffee beans using supercritical CO_2_ extraction with ethanol. Under these optimized conditions, a yield of 6.50% *w*/*w* was obtained, which was accompanied by a high liking score of 7.51. The volatile aroma compounds were determined and found that 3-ethyl-2,5-dimethylpyrazine, 4-vinylguaiacol, 2,5-dimethylpyrazine, furfuryl acetate, furfuryl alcohol, 2,3-butanediol, 5-methyl furfural, 1-furfurylpyrrole, 3-methyl-1,2-cyclopentanedione, 2-phenylethanol, furfural, 2-acetylfuran, 2-pyrrolecarboxaldehyde, 2-acetylpyrrole, pyridine, and thiazole were aroma active compounds in the coffee oil extracted from roasted defective coffee beans. These volatile aroma compounds were associated with roasted, burnt, sweet, bready, and chocolate-like aroma attributes. This study provided information on the application of supercritical CO_2_ with ethanol as an extraction method for coffee oil extraction from roasted defective coffee beans. These findings could show the potential to utilize defective coffee beans as a raw material for coffee oil extraction, as well as utilizing a waste product for the food industry.

## Figures and Tables

**Figure 1 foods-12-02515-f001:**
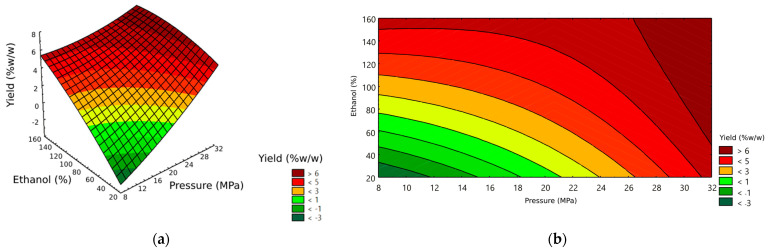
Response surface (**a**) and contour (**b**) plots on yield of coffee oil extraction obtained by supercritical CO_2_ with ethanol extraction (at temperature of 50 °C).

**Table 1 foods-12-02515-t001:** Central composite experimental design for yield (%*w*/*w*) of coffee oil obtained by supercritical CO_2_ with ethanol extraction.

Exp.	Code Values			Actual Values			Yield (%*w*/*w*)
	X_1_	X_2_	X_3_	Pressure (MPa)	Temperature (°C)	Ethanol to Sample Ratio (*w*/*w*)	
1	−1	−1	−1	14.1	40	0.5:1	0.89 ± 0.13
2	1	−1	−1	25.9	40	0.5:1	4.15 ± 0.04
3	−1	1	−1	14.1	60	0.5:1	1.79 ± 0.25
4	1	1	−1	25.9	60	0.5:1	4.06 ± 0.31
5	−1	−1	1	14.1	40	1.19:1	5.42 ± 0.07
6	1	−1	1	25.9	40	1.19:1	6.61 ± 0.23
7	−1	1	1	14.1	60	1.19:1	4.58 ± 0.14
8	1	1	1	25.9	60	1.19:1	5.49 ± 0.09
9	−1.68	0	0	10	50	0.875:1	1.32 ± 0.03
10	1.68	0	0	30	50	0.875:1	7.02 ± 0.21
11	0	−1.68	0	20	33.2	0.875:1	7.68 ± 0.2
12	0	1.68	0	20	66.8	0.875:1	4.81 ± 0.22
13	0	0	−1.68	20	50	0.25:1	1.54 ± 0.19
14	0	0	1.68	20	50	1.5:1	5.14 ± 0.07
15	0	0	0	20	50	0.875:1	3.86 ± 0.07
16	0	0	0	20	50	0.875:1	3.82 ± 0.02
17	0	0	0	20	50	0.875:1	2.63 ± 0.13

**Table 2 foods-12-02515-t002:** ANOVA for the coffee oil extraction yield obtained by supercritical CO_2_ with ethanol extraction.

Source of Variability	Sum of Squares (SS)	Degrees of Freedom (DF)	Mean Square (MS)	F-Value	*p*-Value
Model (regression)	59.53	9	6.61	8.03	0.006
X_1_: Pressure	21.7	1	21.7	26.36	0.001
X_2_: Temperature	2.62	1	2.62	3.18	0.118
X_3_: Ethanol to sample ratio	21.83	1	21.83	26.51	0.001
X_1_X_2_	0.2	1	0.2	0.2429	0.637
X_1_X_3_	1.47	1	1.47	1.79	0.223
X_2_X_3_	0.9647	1	0.9647	1.17	0.315
X_1_^2^	0.2267	1	0.2267	0.2754	0.616
X_2_^2^	8.65	1	8.65	10.51	0.014
X_3_^2^	0.2602	1	0.2602	0.316	0.592
Lack of fit	4.79	5	0.9572	1.96	0.372
Total Error	0.9776	2	0.4888		
Total (corr.)	65.29	16			

X_1_^2^, X_2_^2^, X_3_^2^: Quadratic interaction of pressure, temperature, ethanol to sample ratio; X_1_X_2_: Quadratic interaction between pressure and temperature; X_1_X_3_: Quadratic interaction between pressure and ethanol to sample ratio; X_2_X_3_: Quadratic interaction between temperature and ethanol to sample ratio.

**Table 3 foods-12-02515-t003:** Response factor (*f_i_*) of aroma active components in coffee oil samples.

No.	Compounds	Odor Description	*f_i_*	Equation	R^2^
1	Furfuryl alcohol	bready	1.36	y = 0.737x + 0.195	0.998
2	2,5-Dimethylpyrazine	nutty	1.35	y = 0.739x + 0.141	0.999
3	5-Methyl furfural	burnt, sweet	1.27	y = 0.786x + 0.178	0.998
4	Furfuryl acetate	fruity	1.33	y = 0.750x + 0.204	0.997
5	4-Vinylguaiacol	spicy	1.05	y = 0.954x + 0.402	0.996

**Table 4 foods-12-02515-t004:** Relative concentration (mg/kg, ppm) of key aroma compounds and liking score of coffee oils obtained from various supercritical CO_2_ with ethanol extraction conditions.

Exp.	Concentration (mg/kg, ppm) ^A^						Liking Score ^B^
	Furfuryl Alcohol	2,5-Dimethylpyrazine	5-Methyl Furfural	Furfuryl Acetate	4-Vinylguaiacol	Total	
1	864.48 ^f^ ± 30.70	143.7 ^fg^ ± 3.82	113.8 ^fg^ ± 3.45	66.43 ^e^ ± 1.95	134.3 ^e^ ± 52.63	1322.7 ^h^ ± 28.19	7.29 ^ab^ ± 1.39
2	1240.8 ^ef^ ± 99.69	71.87 ^h^ ± 5.16	106.4 ^fgh^ ± 3.96	85.67 ^e^ ± 1.12	315.5 ^de^ ± 21.99	1820.2 ^gh^ ± 89.71	6.67 ^ab^ ± 1.74
3	1363.5 ^e^ ± 114.5	302.8 ^a^ ± 13.0	362.7 ^a^ ± 3.30	214.0 ^b^ ± 14.2	489.8 ^cde^ ± 11.82	2732.8 ^ef^ ± 126.2	7.14 ^ab^ ± 1.04
4	980.39 ^ef^ ± 49.54	209.2 ^c^ ± 37.6	247.8 ^b^ ± 34.3	158.1 ^c^ ± 65.8	482.2 ^cde^ ± 372.1	2077.7 ^fg^ ± 397.4	7.24 ^ab^ ± 1.56
5	965.27 ^ef^ ± 118.5	154.2 ^efg^ ± 20.0	136.8 ^def^ ± 14.3	83.80 ^e^ ± 7.43	244.1 ^de^ ± 25.87	1584.2 ^gh^ ± 133.8	6.64 ^ab^ ± 1.74
6	1246.9 ^ef^ ± 144.0	121.2 ^g^ ± 14.8	152.7 ^cde^ ± 24.2	135.2 ^cd^ ± 26.7	915.7 ^bcd^ ± 107.4	2571.8 ^ef^ ± 310.4	7.04 ^ab^ ± 1.24
7	2197.8 ^c^ ± 104.1	76.56 ^h^ ± 26.7	79.27 ^h^ ± 29.6	61.65 ^e^ ± 36.3	1104.4 ^bc^ ± 129.9	3519.6 ^cd^ ± 317.0	6.83 ^ab^ ± 1.03
8	1774.0 ^d^ ± 120.4	182.1 ^cde^ ± 6.44	149.9 ^cde^ ± 8.19	87.08 ^e^ ± 5.53	320.7 ^de^ ± 0.370	2513.9 ^ef^ ± 140.7	7.07 ^ab^ ± 1.57
9	1070.0 ^ef^ ± 60.24	289.7 ^a^ ± 3.06	241.1 ^b^ ± 1.58	205.4 ^b^ ± 1.95	400.2 ^cde^ ± 128.1	2206.4 ^fg^ ± 177.5	6.99 ^ab^ ± 1.31
10	1087.0 ^ef^ ± 60.08	158.5 ^def^ ± 31.0	164.3 ^cde^ ± 7.92	205.5 ^b^ ± 48.2	438.8 ^cde^ ± 239.0	2053.5 ^fg^ ± 256.2	7.03 ^ab^ ± 1.67
11	880.33 ^f^ ± 49.02	88.74 ^h^ ± 14.3	85.50 ^gh^ ± 11.4	98.68 ^de^ ± 12.8	186.4 ^de^ ± 85.65	1339.6 ^h^ ± 100.1	6.50 ^b^ ± 1.47
12	1824.1 ^cd^ ± 71.96	194.6 ^cd^ ± 29.1	133.1 ^ef^ ± 19.0	141.7 ^cd^ ± 19.8	384.6 ^cde^ ± 184.6	2678.1 ^ef^ ± 196.1	7.13 ^ab^ ± 1.53
13	1971.3 ^cd^ ± 286.0	255.2 ^b^ ± 35.6	157.0 ^cde^ ± 19.7	264.2 ^a^ ± 26.8	357.6 ^cde^ ± 7.920	3005.3 ^de^ ± 239.6	7.27 ^ab^ ± 1.41
14	2025.2 ^cd^ ± 336.9	191.4 ^cd^ ± 2.52	165.1 ^cde^ ± 29.1	150.1 ^c^ ± 22.7	1334.9 ^b^ ± 67.19	3866.7 ^c^ ± 245.4	7.17 ^ab^ ± 1.53
15	3546.8 ^a^ ± 403.8	211.3 ^c^ ± 5.98	181.0 ^c^ ± 19.0	159.4 ^c^ ± 0.27	499.7 ^cde^ ± 329.9	4598.2 ^b^ ± 751.4	6.92 ^ab^ ± 1.17
16	3698.4 ^a^ ± 134.2	188.0 ^cde^ ± 3.04	160.7 ^cde^ ± 13.0	143.4 ^cd^ ± 8.71	1510.9 ^b^ ± 409.6	5701.5 ^a^ ± 371.8	7.06 ^ab^ ± 1.63
17	2590.7 ^b^ ± 568.2	191.0 ^cd^ ± 5.84	168.8 ^cd^ ± 8.72	148.0 ^c^ ± 9.80	2314.7 ^a^ ± 1413.2	5413.2 ^a^ ± 937.4	7.47 ^a^ ± 1.02

Means ± standard deviation (^A^ *n* = 3 and ^B^ *n* = 30) with different letters within a column are significantly different (*p* ≤ 0.05).

**Table 5 foods-12-02515-t005:** Concentration of key aroma compounds of coffee oil at optimum condition.

Compounds	Concentration (mg/kg, ppm)
Furfuryl alcohol	2363.02 ± 393.29
2,5-Dimethylpyrazine	208.61 ± 41.22
5-Methyl furfural	217.03 ± 49.84
Furfuryl acetate	160.63 ± 48.1
4-Vinylguaiacol	203.05 ± 2.13
Total	3152.34 ± 527.2

Means ± standard deviation (*n* = 3).

**Table 6 foods-12-02515-t006:** Volatile aroma compounds in coffee oil by the direct solvent extraction (DSE).

No.	RI ^A^	Ref. RI ^B^	Compound	Relative Concentration ^C^ (mg/kg, ppm)	Odor Threshold ^D^ (mg/kg, ppm)	OAV	Odor Description	Identification ^E^
1	<1100	n.a.	Unknown	n.d.	n.a.	n.a.	fruity	O
2	<1100	n.a.	Unknown	n.d.	n.a.	n.a.	sweet	O
3	1208	1194 ^I^	Pyridine	52.43 ± 19.86	0.1	524	burnt	MS,RI,O
4	1275	1269 ^I^	Thiazole	44.09 ± 16.19	0.038	1160	meaty	MS,RI,O
5	1363	1348 ^I^	*2,5-Dimethylpyrazine	208.61 ± 41.22	0.08	1932	nutty, chocolate	MS,RI,O
6	1419	1410 ^I^	Furfural	75.46 ± 17.12	0.77	98	bready	MS,RI,O
7	1424	1454 ^II^	Acetoxyacetone	74.72 ± 28.65	n.a.	n.a.	fruity	MS,RI,O
8	1466	1467 ^I^	2-Acetylfuran	52.93 ± 18.94	10	5	nutty	MS,RI,O
9	1499	n.a.	Unknown	n.d.	n.a.	n.a.	fermented	O
10	1507	1512 ^I^	*Furfuryl acetate	160.63 ± 48.1	0.1	1208	fruity	MS,RI,O
11	1510	n.a.	Unknown	n.d.	n.a.	n.a.	meaty	O
12	1512	1480 ^I^	3-Ethyl-2,5-dimethylpyrazine	63.5 ± 27.66	0.0004	158,738	burnt, sweet	MS,RI,O
13	1528	1546 ^I^	2,3-Butanediol	49.09 ± 21.44	0.0951	516	balsamic, sweet	MS,RI,O
14	1536	1539 ^I^	*5-Methyl furfural	217.03 ± 49.84	0.5	342	burnt, sweet	MS,RI,O
15	1560	1613 ^I^	*Furfuryl alcohol	2363.02 ± 393.29	1.9	914	bready	MS,RI,O
16	1811	1831 ^I^	3-Methyl-1,2-cyclopentanedione	55.12 ± 13.98	0.3	184	burnt, sweet	MS,RI,O
17	1822	1837 ^I^	1-Furfurylpyrrole	30.71 ± 9.5	0.1	307	burnt, bready	MS,RI,O
18	1829	n.a.	Unknown	n.d.	n.a.	n.a.	spicy	O
19	1834	1857 ^I^	2-Phenylethanol	43.79 ± 5.38	0.39	112	floral	MS,RI,O
20	1930	1978 ^I^	2-Pyrrolecarboxaldehyde	56.23 ± 17.91	65	1	musty	MS,RI,O
21	1941	n.a.	Unknown	n.d.	n.a.	n.a.	burnt	O
22	1959	1927 ^I^	2-Acetylpyrrole	86.92 ± 29.45	58.58	1	woody, musty	MS,RI,O
23	2116	2146^I^	*4-Vinylguaiacol	203.05 ± 2.13	0.019	10,178	phenolic, spicy	MS,RI,O

^A^ RI (wax): Experimental retention index on an DB-Wax column relative to C10–C40 alkane standards. ^B^ Ref. RI: Reference retention index values from literature: ^I^ NIST library version 2.2, ^II^ Caporaso et al. [36]. ^C^ The relative concentration of compounds which without (*) was determined against peak area of each volatile to the internal standard (2,4,6-trimethylpyridine) by assuming all response factors = 1. ^D^ Odor threshold values in water (mg/kg, ppm) from Van Gemert. ^E^ Identification methods: MS = mass spectra; RI = retention index; O = odor description. n.d. is not detected, n.a. is not available.

## Data Availability

Data are contained within the article.

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
