# Peer review of "Optimization of Coffee Oil Extraction from Defective Beans Using a Supercritical Carbon Dioxide Technique: Its Effect on Volatile Aroma Components"

_foods, 2023, doi:10.3390/foods12132515_

Round 1
Reviewer 1 Report
Q1:The effect of different extraction processes on the flavour of coffee oil is one of the main focuses of this paper, and there is little description in the text of the differences in flavour of coffee oil extracted by different processes, so please add to this.
Q2:Lines 22-23 of the abstract say "different oil yields with different volatiles aroma compounds concentrations were obtained", What is the exact relationship between "oil yields" and "volatiles aroma compounds concentrations were obtained", which is not mentioned later.。
Q3:The results of the quantitative analysis of several flavour substances can be added to the description of the flavour in lines 25-27 of the abstract.
Q4:The main contributions and significance of the study are not highlighted and may be supplemented by appropriate descriptions.
Q5:Table 4 shows the relative concentrations of the coffee oil aroma compounds, but the table headings are not clearly expressed.
Q6:To the conclusions can be added the main contributions of the various aroma compounds respectively in coffee oils.
Q7:Overall, there are more tables throughout the text and some data can be analysed and compared by making graphs.
Author Response
|
Reviewer’s comments |
Author’s comments and changes |
|
1. The effect of different extraction processes on the flavour of coffee oil is one of the main focuses of this paper, and there is little description in the text of the differences in flavour of coffee oil extracted by different processes, so please add to this. |
Author added more information. Revised as suggested. Please see Line 76-88. |
|
2. Lines 22-23 of the abstract say "different oil yields with different volatiles aroma compounds concentrations were obtained", What is the exact relationship between "oil yields" and "volatiles aroma compounds concentrations were obtained", which is not mentioned later. |
Author added more information. Revised as suggested. Please see Line 362-369. |
|
3. The results of the quantitative analysis of several flavour substances can be added to the description of the flavour in lines 25-27 of the abstract.
|
Revised as suggested. Please see Line 26-28. |
|
4. The main contributions and significance of the study are not highlighted and may be supplemented by appropriate descriptions. |
Author added more information. Revised as suggested. Please see Line 369-374. |
|
5. Table 4 shows the relative concentrations of the coffee oil aroma compounds, but the table headings are not clearly expressed.
|
Revised as suggested. Please see Line 348-349. |
|
6. To the conclusions can be added the main contributions of the various aroma compounds respectively in coffee oils. |
Revised as suggested. Please see Line 511-514. |
|
7. Overall, there are more tables throughout the text and some data can be analysed and compared by making graphs. |
Thank you for the comment. We tried to convert data in the Table 4 to be graphs; however, it turns out that displaying data in table format would be more appropriate. We will take this consideration for further studies. |
Reviewer 2 Report
This article is devoted to the process of numerical optimization of the extraction of coffee oil from defective beans by the method of supercritical carbon dioxide. Also in this work, an assessment of its effect on volatile aromatic components was carried out. The article is written in an accessible language and well structured, the main ideas are not in doubt. The article is relevant, since obtaining additional volatile components from defective coffee beans has a number of prospects. The article fits the requirements of the journal in terms of volume and subject matter. There are a number of points that need to be improved:
1. Why did the authors choose green arabica coffee beans?
2. Why did the authors choose this particular model for calculations? Why was the Box-Behnken model not used?
3. How were confidence intervals calculated?
4. How many parallel experiments were done?
5. Was the confirmatory experiment carried out under optimal (calculated) conditions?
6. In addition, it is desirable to add more comparison of the obtained data with literature sources.
7. Please cite: 10.3390/molecules27186129.
8. What is the reliability of the obtained mathematical model?
9. It is desirable to expand the conclusions.
Author Response
|
Reviewer’s comments |
Author’s comments and changes |
|
1. Why did the authors choose green arabica coffee beans? |
We are interested in arabica coffee since it was reported that arabica provided more refine and aromatic in terms of aroma and flavor. Thus, we hypothesize that the defective green arabica coffee could produce roasted coffee oils that can provide a better aroma/flavor than the Robusta. In addition, from the literatures, it was found that the arabica species exhibited a higher concentration of volatile aroma compounds than Robusta species, such as furfural, 2,5-dimethylpyrazine, 5-methyl furfural, etc. (Caporaso, 2018). These compounds are reported as impact compounds of the roasted coffee aroma. Moreover, the lipids content of arabica was higher (15-18%) than that found in Robusta (8-12%) (Belitz et al., 2009). From above reasons, the green arabica coffee was selected for the study. |
|
2. Why did the authors choose this particular model for calculations? Why was the Box-Behnken model not used? |
The central composite design (CCD) was applied for experimental design due to its specific advantages over the Box-Behnken model. The CCD model offers a higher degree of flexibility in experimental design, providing more comprehensive assessment of the response surface (+α,-α). Therefore, the CCD model was applied to gain a more thorough understanding of the factors and their interactions influencing the response variables instead of using the Box-Behnken model. |
|
3. How were confidence intervals calculated? |
Two-way analysis of variance (ANOVA) and Duncan's new multiple range tests (DNMRT) were employed to analyze differences in the mean values at a 95% confidence interval. It was added in the manuscript. Please see Line 215-217. |
|
4. How many parallel experiments were done? |
-All measurements were performed in triplicate experiments. It was added in the manuscript. Please see Line 219-220. |
|
5. Was the confirmatory experiment carried out under optimal (calculated) conditions? |
The validation of the optimal condition obtained from the calculation was done in triplicates. The optimized condition was chosen for validation to confirm results, which were employed from triplicated experiments. It was added in the manuscript. Please see Line 292-294. |
|
6. In addition, it is desirable to add more comparison of the obtained data with literature sources. |
Author added more information. Revised as suggested. Please see Line 233-235 and 443-446. |
|
7. Please cite: 10.3390/molecules27186129. |
Author added more information. Revised as suggested. Please see Line 60-62. And added as Reference no.12. |
|
8. What is the reliability of the obtained mathematical model? |
- The information of regression model was added into the Table 2. It was shown that the regression model was significant (p<0.05). Also, the coefficient of determination (R2) of the model was 0.912, which indicated that the model can explain 91.2% of the total variation in the % yield equation. This high R2 value could suggest that the obtained mathematical model is reliable and provides a good fit to the data. |
|
9. It is desirable to expand the conclusions. |
Revised as suggested. Please see Line 511-516. |
Reviewer 3 Report
Here are some comments and suggestions for the abstract:
- The abstract provides a clear overview of the study, including the objectives, methodology, and key findings. It effectively summarizes the main points of the research.
- Consider providing more specific information about the types of defects in the coffee beans that can produce volatile aroma compounds after roasting. This would help readers understand the relevance and potential applications of the study.
- It would be helpful to mention the significance of optimizing coffee oil extraction from defective beans. Highlighting the potential benefits, such as reducing waste and utilizing resources more efficiently, could enhance the importance of the research.
- Specify the range of the optimized operational temperatures, pressure, and ethanol to roasted coffee ratio in a more concise manner. Instead of stating the full range, you could mention the specific values chosen for the optimization process.
- When discussing the obtained results, provide more context regarding the differences in oil yields and concentrations of volatile aroma compounds under different extraction conditions. This would help readers understand the implications of these variations and their potential impact on the sensory quality of the extracted coffee oil.
- Consider expanding on the potential applications of the coffee oil extracted from defective beans as a flavoring agent. Discuss the specific industries or products that could benefit from using this oil, such as the food and beverage industry or the development of new coffee-based products.
- It would be helpful to mention any limitations or future directions for the research. This could include areas of further investigation, such as studying the stability or shelf life of the extracted coffee oil or conducting sensory evaluations to validate its suitability as a flavoring agent.
- Ensure that the abstract is structured in a clear and concise manner, with each paragraph addressing a specific aspect of the study (e.g., objectives, methodology, results, implications).
Overall, the abstract provides a good summary of the study. By addressing the suggestions above, you can further enhance its clarity and impact.
no issues
Author Response
|
Reviewer’s comments |
Author’s comments and changes |
|
1. Consider providing more specific information about the types of defects in the coffee beans that can produce volatile aroma compounds after roasting. This would help readers understand the relevance and potential applications of the study. |
Author added more information. Revised as suggested. Please see Line 49-57. |
|
2. It would be helpful to mention the significance of optimizing coffee oil extraction from defective beans. Highlighting the potential benefits, such as reducing waste and utilizing resources more efficiently, could enhance the importance of the research. |
Author added more information. Revised as suggested. Please see Line 117-119 and 512-516. |
|
3. Specify the range of the optimized operational temperatures, pressure, and ethanol to roasted coffee ratio in a more concise manner. Instead of stating the full range, you could mention the specific values chosen for the optimization process. |
Revised as suggested. Please see Line 149-150, and 290-294. |
|
4. When discussing the obtained results, provide more context regarding the differences in oil yields and concentrations of volatile aroma compounds under different extraction conditions. This would help readers understand the implications of these variations and their potential impact on the sensory quality of the extracted coffee oil. |
Author added more information. Revised as suggested. Please see Line 362-369. |
|
5. Consider expanding on the potential applications of the coffee oil extracted from defective beans as a flavoring agent. Discuss the specific industries or products that could benefit from using this oil, such as the food and beverage industry or the development of new coffee-based products. |
Author added more information. Revised as suggested. Please see Line 495-498. |
|
6. It would be helpful to mention any limitations or future directions for the research. This could include areas of further investigation, such as studying the stability or shelf life of the extracted coffee oil or conducting sensory evaluations to validate its suitability as a flavoring agent. |
Author added more information. Revised as suggested. Please see Line 498-500. |
|
7. Ensure that the abstract is structured in a clear and concise manner, with each paragraph addressing a specific aspect of the study (e.g., objectives, methodology, results, implications). |
Revised as suggested. |